# The association of herd performance indicators with dairy cow longevity: An empirical study

**Ruozhu Han**[1]*, **Monique Mourits**[1], **Wilma Steeneveld**[2], **Henk Hogeveen**[1]

**1** Department of Social Sciences, Business Economics Group, Wageningen University, Wageningen, The Netherlands, **2** Faculty of Veterinary Medicine, Department of Population Health Sciences, Utrecht University, Utrecht, The Netherlands

ⓥ These authors contributed equally to this work.

* ruozhu.han@wur.nl

**Data Availability Statement:** For detailed information on the data management procedures by which the original herd data are processed and stored, please see CRV's privacy statement (https://crv4all.com/en/privacy-statement).

## Abstract

The associations between reproductive performance, milk yield and health status with the risk of culling, and thus with a cow's longevity, have been well documented at the individual cow level. Associations at individual cow level may, however, not be valid at herd level due to interrelated herd management aspects and/or policy restrictions. The objective of this study was to explore the association of herd performance indicators with herd-level dairy cow longevity under Dutch production conditions. Longevity was expressed by three different measures, viz. age at culling, lifetime milk production of culled cows and culling rate. The evaluated herd performance indicators included factors on milk production, youngstock rearing, reproduction and health performance as registered on 10 719 Dutch commercial dairy herds during the period 2007–2016. Averaged over herds and the evaluated period, the age of culled milking cows was 2 139 days (5.8 years, SD±298 days), the lifetime milk production of culled cows was 31 238 kg (SD±7,494 kg), and the culling rate was 0.24 (SD ±0.08). A mixed linear regression modelling approach was applied to evaluate the association of each of the three longevity measures with the selected herd performance indicators. The results indicated that only four herd performance indictors (herd size, herd expansion, heifer ratio and the proportion of cows with potential subclinical ketosis) shared significant associations with all three longevity variables. Generally, the strength of the associations between each of the evaluated longevity measures and herd performance indicators was only limited. The absence of strong associations between the longevity measures and herd performance indicators reveal that there is potential of extending cattle longevity without affecting the herd performance in terms of milk production, reproduction and health. Moreover, only part of the observed variance in longevity among the herds over time was explained by the herd performance variables, indicating that differences in longevity at herd level may predominantly be determined by other factors, like farmers' attitude and strategic management.

Requests to access the original data need to be directed to CRV (https://crv4all.com/). Contact for data access and ethics committeeL Gerben de Jong (gerben.de.jong@crv4all.com).

**Funding:** This study was financially supported by the Sino-Dutch Dairy Development Centre(SDDDC) and China Scholarship Council (CSC). The funders had no role in study design, data collection and analysis, decision to publish, or preparation of the manuscript.

**Competing interests:** The authors have declared that no competing interests exist.

## Introduction

Dairy cattle longevity can be defined as the duration of the life of a dairy cow, reflected by the time from birth until the moment of culling from the herd [1]. In the Netherlands, the lifespan of milking dairy cows in a herd is on average about 5.8 years ranging from 4.9 years to 7.1 years [2]. This is, however, below the potential natural lifespan of dairy cows. Prolonging dairy cattle longevity is one of the potential options to contribute to more sustainable milk production, from an economic, an environmental as well as a social perspective [1]. In the Netherlands, a farmer's decision to cull a dairy cow is largely driven by economic considerations [3], by comparing the expected performance of the present cows (defined by its observed productive performance, reproduction and health status) against the expected future performance of the available replacement animals. As long as a prolonged longevity does not trigger higher risks of reproductive problems, health disorders or both, increased longevity could improve economic efficiency by reducing rearing costs, as fewer replacement heifers are needed, and increasing the average herd production [4].

Dairy production is an important source of greenhouse gas (GHG) emissions [5]. An increased longevity could contribute to a more environmental-friendly milk production by reducing the GHG emissions from youngstock rearing. Moreover, an increased longevity is perceived by society as a relevant indicator of animal welfare [6]. Low culling rates, especially low mortality rates, indicate that a herd keeps a large group of existing cows alive and has the ability to maintain expected production. Up till now, culling decisions were purely regarded as the responsibility of the farmer. However, because of the increased general interest in environment (i.e., GHG emissions) and animal welfare (i.e., the moral concerns regarding premature culling of dairy cows), there is an increased societal interest for a prolonged dairy cattle longevity. To align with these societal interests, Dutch dairy processing companies and farmers' associations actively advocate prolongation of cattle longevity [7].

Longevity can be expressed in several ways. Besides average lifespan or average age at culling, other measures are used to express longevity, like average lifetime milk production and culling rate [1]. Unlike age at culling, lifetime milk production not only depends on the average lifespan, but also on milk production intensity. Therefore, lifetime milk production measured as the kg of milk produced by the cow during its lifespan embodies a more economic and environmental perspective on longevity [8]. As indicated earlier, animal welfare is often assessed by society on the basis of the proportion of culled cows in the herd, making the culling rate a relevant longevity feature from a societal perspective.

A considerable number of studies have been carried out on the risk factors that are associated with culling or longevity on cow level. In these studies, poor fertility and health disorders, especially the occurrence of mastitis, have been indicated as the most important risk factors to cow survival (e.g., [9]). Significant associations between reproductive traits and functional longevity on cow level have been demonstrated by Pfeiffer et al [10] and Sewalem et al [11]. In addition, significant impacts of low productivity on the risk of an individual cow to be culled have been verified as well [12]. These associations at individual cow level may, however, differ at herd level; partially because of interrelated herd management aspects (e.g. young stock rearing) [13] and herd size restrictions imposed by policy regulations like milk quota before 2015 or phosphate restrictions after 2018 [1]. For instance, a cow with a relatively low production has a higher probability of being culled in comparison to a high productive cow within the same herd, while a herd with a relatively low average milk production may not have a higher culling rate than a herd with a high average milk production. Scientific literature on the association of herd performance indicators with herd-level longevity performance is lacking. Only a few studies so far have explored the association of longevity with herd performance indicators

such as herd size, herd turnover rate, and milk yield by the use of empirical data (e.g., [14,15]). More extensive insights at herd level are needed when targeting farmer and policy maker towards an increase in dairy cow longevity.

The objective of this study was to explore the association of herd performance indicators with different herd-level dairy cow longevity aspects (age at culling, lifetime milk production and culling rate) using Dutch production conditions. The herd performance indicators included factors on milk production, youngstock rearing, reproduction and health performance as derived from Dutch dairy herd data registered during the period 2007–2016.

## Materials and methods

### Available data

Data over the period 2007–2017 were provided by the Dutch Cooperative Cattle Improvement Organization CRV BV (CRV, Arnhem, the Netherlands) with consent of their associated farmers. To prevent researchers to see any information about individual farmers, a separate, independent, company (iDTS, the Netherlands) did anonymize the data so that data could not be traced back to individual farmers. A contract between the data provider, the data anonymizing company and the authors' university (Wageningen University) ensures the correct management procedures of herd data. The data included information on 20 796 dairy herds (mainly Holstein-Friesian), representing approximately 80 percent of all dairy herds in the Netherlands. The dataset contained herd data regarding longevity features, reproduction performance, milk production performance and health status. Information on herd longevity features were available as annual averages for the production year (September-August). Reproduction performance features were registered by annual calendar year averages (January—December). Data on herd milk production performance were routinely collected on several occasions per year (i.e., test days with intervals of 3–9 wks). These milk production data were averaged over the multiple test days in one calendar year to obtain a yearly herd average. The information on the herd health status was derived from individual cow level data and averaged per calendar year at herd level. From the overall database, 19 variables were selected for further data editing and analyzes based on their expected relation with longevity [16,17]. An overview of the selected variables is displayed in Table 1.

This observational study uses anonymized dairy herd management data that is originally collected, stored and processed by CRV according to their privacy statement on the use of personal data. Within this statement dairy farmers are informed about the purposes for which the collected data could be used (like research) and under what conditions and about their consent and rights to object in this respect. As the data had been anonymized by a third independent party prior to access and analysis, and the study did not pose any potential risks to individual dairy farmers or their privacy, no additional consent was requested for the use of the data in this study.

### Data editing

The study focused on commercial Dutch dairy herds, which remained in production throughout the evaluated period of 2007–2017. Until 2017, 20 591 herds met the condition of successive farming. Due to the lack of data after 2017 to prove continued farming thereafter, the data entries from 2017 were excluded from database. To classify a herd as a commercially viable dairy herd, a herd had to adhere to four conditions. Firstly, they had to have at least 6 and not more than 17 test days per calendar year (excluding 7 857 herds). Secondly, the number of milking cows within a herd should not have been less than 30 in each calendar year (excluding 1 167 herds). Thirdly, the herd-level average milk production per 305 days must have been

**Table 1. Descriptive statistics on the selected herd performance indicators, indicating the distribution of herd averaged values across herds and years (2007–2016).**

| Variables | Description | | Records, N | Mean | SD | Percentile _5[5] | Percentile _95[5] |
|---|---|---|---|---|---|---|---|
| Longevity | | | | | | | |
| Age culled cows | Age of culled milking cows (days) | | 107 176 | 2 139 | 298 | 1 719 | 2 674 |
| Life. prod. culled cows | Lifetime milk production of culled milking cows (kg) | | 107 177 | 31 238 | 7 495 | 20 344 | 44 465 |
| Culling rate[1] | Annual rate of culled milking cows over number of milking cows | | 107 168 | 0.24 | 0.08 | 0.12 | 0.38 |
| Herd structure | | | | | | | |
| Herd size | Number of milking cows | < = 50 (Small) | 12 034 | 43 | 5 | 34 | 50 |
| | | 51–100 (Medium) | 62 680 | 74 | 14 | 53 | 97 |
| | | 101–200 (Large) | 29 413 | 129 | 24 | 102 | 181 |
| | | >200 (Very large) | 3 063 | 267 | 90 | 203 | 432 |
| Herd expansion[1] | Rate of herd size change relative to herd size of 2007 | | 107 190 | 1.15 | 0.25 | 0.91 | 1.61 |
| Youngstock | | | | | | | |
| Heifer ratio[1] | Ratio of first calving heifers over number of milking cows | | 107 176 | 0.24 | 0.06 | 0.14 | 0.34 |
| Birth first AI | Interval between birth and first AI (days) | | 88 912 | 491 | 54 | 433 | 588 |
| Calve born first AI youngstock[2] | Percentage of calves born after the first AI of youngstock | | 104 870 | 52.48 | 18.15 | 22.22 | 81.82 |
| N AI success youngstock[2,3] | Number of AIs until conception per head of pregnant youngstock | | 98 429 | 1.8 | 0.4 | 1.1 | 2.6 |
| N AI youngstock [2] | Number of AIs per head of inseminated youngstock | | 103 558 | 1.9 | 0.5 | 1.1 | 2.8 |
| Age first calving | Age at the first calving (days) | | 105 731 | 792 | 48 | 733 | 881 |
| Reproduction | | | | | | | |
| CI[3] | Calving interval (days) | | 105 427 | 415 | 26 | 380 | 462 |
| Calving first AI[3] | Interval from calving to first AI (days) | | 103 980 | 99 | 24 | 72 | 140 |
| Calve born first AI[4] | Percentage of calves born after the first AI after calving | | 106 070 | 49.73 | 15.27 | 26.39 | 73.40 |
| N AI success[2,4] | Number of AIs until conception per head of pregnant dairy cow | | 98 633 | 1.8 | 0.4 | 1.2 | 2.5 |
| N AI[2,4] | Number of inseminations per dairy cow | | 104 298 | 2.0 | 0.5 | 1.2 | 2.8 |
| Health status | | | | | | | |
| High SCC | Percentage of primiparous cows with SSC over 150,000 cells/ml and multiparous cows with SCC over 200,000 cells/ml over average number of milking cows | | 102 301 | 19.49 | 7.13 | 9.12 | 32.20 |
| Newhigh SCC | Percentage of new cows in high SCC | | 102 300 | 8.89 | 2.85 | 4.87 | 13.54 |
| SCC | Number of somatic cells per ml of milk *1,000 (cells/ml) | | 102 205 | 202 | 63 | 112 | 318 |
| Suspect subclinical ketosis | Ratio of suspected cows with subclinical ketosis over average number of milking cows | | 107 190 | 0.04 | 0.02 | 0.01 | 0.07 |
| Milk production | | | | | | | |
| FPCM305[1] | Fat protein corrected milk production within 305 days of lactation (kg) | | 106 665 | 9 093 | 900 | 7 534 | 10 527 |

[1] Derived indicators, computed by variables registered by CRV.

[2–4] Removed indicator due to P-value of univariate analysis<0.25 in model of age of culled milking cows, lifetime milk production of culled milking cows and culling rate, respectively.

[5] The value of variable in 5 or 95 percentile.

over 6 000 kg per calendar year (excluding 359 herds). Finally, herds should have had an annual culling rate between 0.1 and 0.5 for 8 out of the 10 calendar years (excluding 431 herds). The final dataset on herd performance consisted of 10 719 herds over the years 2007–2016.

Subsequently, data entries reflecting biological unrealistic values were removed from the database. Nine entries were removed because of annual culling rates ≥1. From the herd reproduction performance data, 143 entries were removed due to average calving intervals (CI) on herd level < 310 days or > 600 days. Also, entries indicating a herd average interval between calving to first artificial insemination (AI) of milking cows < 30 days or > 365 days were removed (142 entries). Twenty entries with an average age at first calving < 500 days or > 1 500 days were removed. Entries indicating the ratio of number of the first calving heifer over the milking cow ≥1 were removed (14 entries), Lastly, one entry was removed because of an annual percentage of new cows with a high somatic cell count (SCC) in a herd ≥100%.

**Cattle longevity variables.** Three annually (production year) averaged cattle longevity variables were selected: age of culled milking cows (days), lifetime milk production of culled milking cows (kg), and culling rate. The number of culled cows represented all dairy cows, after first calving, removed from the milking herd for slaughter, salvage or death within a production year, following the definition used by Fetrow et al [18]. Animals sold for production purposes to other dairy farms were excluded from this number. The annual average culling rate was computed by dividing the number of culled milking cows by the number of milking cows in the same production year.

**Herd size variables.** The herd size was represented by the number of milking cows within a herd on the end of August which was stratified into 4 groups, representing small, medium, large and very large sized herds. The stratification thresholds for these categories were 50, 100, 200 cows, respectively. To account for herd expansion during the evaluated period, a herd expansion ratio was derived by relating the sizes of a herd through time to its reference herd size of 2007.

**Young stock variables.** The young stock performance indicator heifer ratio was derived by dividing the number of first calving heifers by the number of milking cows in a production year. Reproduction performance of young stock was presented by the annual averaged herd data on the interval between birth to first AI, percentage of calves born after the first AI, number of AIs until conception, number of AIs and age at first calving.

**Reproduction variables.** The herd reproduction performance of milking cows was reflected by the annual averaged herd data on CI, interval between calving to first AI, percentage of calves born after the first AI, number of AIs until conception and number of AIs.

**Health variables.** Data that reflected the herd health status in relation to subclinical mastitis included average SCC, the annual percentage of cows with a high SCC and the annual percentage of new cows with a high SCC in a herd. SCC over 150 000 cells/ml for primiparous cows and over 200 000 cells/ml for multiparous cows have been used to define high SCC counts. These thresholds are used in the Netherlands as a standard to reflect subclinical mastitis [19,20]. The percentage of cows with a high SCC and new cows with a high SCC were collected on each test day and averaged over all test days within a calendar year.

To get insight in the metabolic status of a herd, the fat protein ratio during the first 100 DIM was used as an indicator, where a ratio > 1.5 was regarded as an indication of the occurrence of subclinical ketosis [21]. Given this ratio per cow per test day, the herd ratio of cows with a high fat protein ratio was determined. The average ratio of suspected cows with subclinical ketosis within a herd was calculated as an average over all test days within one calendar year.

**Milk production variable.** Based on the 305-day milk (kg), fat (%) and protein (%) production, the fat and protein corrected milk production in 305 days (FPCM305) was

determined as follows [22]:

$$
\begin{aligned}
\text{FPCM305(kg)} &= \text{milk prod305days(kg)} * (0.337 + 0.116 * \text{Fat content 305 days (\%)} + 0.06 \\
&\quad * \text{Protein content 305 days(\%)})
\end{aligned}
$$

## Data analyses

A mixed linear regression model was applied to analyze the association of herd performance indicators with the three herd-level longevity variables, i.e., age of culled milking cows, lifetime milk production of culled milking cows and annual culling rate. Herd as random effect was included in the linear mixed model to capture any other unobserved herd heterogeneity, such as specific herd management. To correct for repeated measurements, a covariance structure was adopted based on the Akaike information criterion (AIC). Competing covariance structures (i.e., independent, compound symmetry, first-order autoregressive and unstructured) were tested for their fit and the structure with the lowest AIC, the unstructured covariance structure, was regarded to give the best fit and chosen for the final modelling. A year variable was forced into all models to capture potential time effects (e.g., milk price changes). Explanatory variables were selected based on the following five steps: (1) a linear relationship check between each pair of explanatory variable and longevity variable; (2) continuous variable's normality distribution check; (3) univariate analysis; (4) collinearity screening; and (5) stepwise regression. All data manipulations and modelling were performed in R [23].

**Linear relationship check.** Continuous variable normality check and univariate analysis. The linearity of the relationship between the selected explanatory variables and longevity variables was visually inspected by creating boxplots. Histograms and descriptive statistics were used to scrutinize the distribution of continuous variables and to assist any further categorization or transformation. As a result, seven skewed variables were log transformed to be more normally distributed and to stabilize the variance, including the variable of herd expansion, percentage of new cows with high SCC in a herd, CI, interval between calving to first AI, interval between birth to first AI, number of AIs until conception and age at first calving. For all potential explanatory values, a univariate analysis with each of the three longevity variables was carried out. Variables with a P-value <0.25 were kept for the final multivariable modelling process [24]. Consequently, five, three and three variables were removed from the age and lifetime milk production of culled milking cows and culling rate models, respectively (see Table 1).

**(Multi)collinearity.** For the remaining variables, potential collinearity was identified by examining the Pearson correlation coefficients, where a correlation coefficient larger than or equal to 0.8 was used to determine collinearity. In addition, independent variables with a variance inflation factor of more than or equal to 10 were considered as variables causing serious multicollinearity problems [25]. For this reason, two potential explanatory variables were removed from all three longevity models: i.e., average SCC and average percentage of new cows with high SCC (highly correlated with average percentage of cows with high SCC). Moreover, in the model for lifetime milk production of culled milking cows, the variable average number of AIs of cows until conception was omitted (highly correlated with average number of AIs of cows). In the model on culling rate, the variable average number of young stock AIs until conception (highly correlated with average number of young stock AIs) was excluded.

**Stepwise regression.** The herd indictors were tested using a backward stepwise selection procedure. This procedure continued until the marginal change in the Bayesian information criterion (BIC) between selection steps increased rapidly.

In order to compare the explanatory value of the regressed performance indicators on longevity, the continuous explanatory variables were, subsequently, transformed by centering and scaling to obtain insight in the change in estimated longevity value resulting from one standard deviation increase of in explanatory variable value.

## Results

### Longevity descriptive statistics

Averaged over the 10 719 herds and the 10-yrs period, the age of culled milking cows in a herd was 2 139 days (5.8 years, SD±298 days) (Table 1). The lifetime milk production of culled milking cows in a herd was 31 238 kg (SD±7,494 kg), and the culling rate in a herd was 0.24 (SD ±0.08). The three defined longevity variables had a right-skewed distribution. Although the annually averaged age and lifetime milk production of culled milking cows did not alter much throughout the evaluated 10 years, the variance around these two average variables decreased over the years. This is in contrast to the variance of the annual culling rate, which showed no tendency of change over time. (Table 2).

Cattle longevity varied with herd size (Table 2). Average age of culled milking cows decreased with increasing herd size (trend coefficient P<0.001); the average age at culling in the smallest sized herds (<50 cows) was 130 days higher than in the largest sized herds (>201 cows). Average lifetime milk production of culled milking cows reached a plateau for herds having a medium herd size (51~100 cows). The very large herd size (>201 cows) had the lowest average lifetime milk production of 30 127 kg. Moreover, the standard deviation of these two longevity variables decreased from small (<50 cows) to very large (>201 cows) herds. The average culling rate in a herd slightly varied among herd sizes from 0.24 to 0.26. Although the

**Table 2. Herd averages on age of culled milking cows (days), lifetime milk production of culled cows (kg) and culling rate with SD by year and herd size.**

| | Age culled cows (days) | | Lifetime production culled cows (kg) | | Culling rate | |
|---|---|---|---|---|---|---|
| | Mean | SD | Mean | SD | Mean | SD |
| Year | | | | | | |
| 2007 | 2 146 | 318 | 30 523 | 7 952 | 0.22 | 0.09 |
| 2008 | 2 171 | 319 | 31 286 | 7 970 | 0.21 | 0.08 |
| 2009 | 2 160 | 303 | 31 229 | 7 492 | 0.26 | 0.09 |
| 2010 | 2 141 | 296 | 31 125 | 7 327 | 0.25 | 0.09 |
| 2011 | 2 119 | 286 | 30 938 | 7 226 | 0.26 | 0.09 |
| 2012 | 2 121 | 298 | 31 205 | 7 448 | 0.24 | 0.08 |
| 2013 | 2 130 | 296 | 31 338 | 7 569 | 0.22 | 0.07 |
| 2014 | 2 138 | 289 | 31 562 | 7 315 | 0.25 | 0.08 |
| 2015 | 2 131 | 292 | 31 493 | 7 371 | 0.23 | 0.08 |
| 2016 | 2 135 | 280 | 31 683 | 7 169 | 0.24 | 0.08 |
| Herd size[1,2] | | | | | | |
| Small | 2 191 | 370 | 31 107 | 9 025 | 0.26 | 0.26 |
| Medium | 2 149 | 304 | 31 395 | 7 657 | 0.24 | 0.08 |
| Large | 2 105 | 252 | 31 074 | 6 536 | 0.23 | 0.07 |
| Very large | 2 061 | 212 | 30 127 | 5 822 | 0.24 | 0.07 |

[1] Herd size was stratified by small, medium, large and very large based on the number of milking cows > = 50 (2269 herds), 51–100 (8515 herds), 101–200 (5003 herds) and > = 201 cows (682 herds), respectively.

[2] The significant associations (p<0.001) tested by simple linear regression.

**Table 3. Results of the mixed linear regression models on the association of age and lifetime milk production of culled milking cows and culling rate with herd performance indicators.**

| | | Age culled cows (days) | | | Lifetime production culled cows (kg) | | | Culling rate | | |
|---|---|---|---|---|---|---|---|---|---|---|
| | | Regression coefficient | SE | P-value | Regression coefficient | SE | P-value | Regression coefficient | SE | P-value |
| Intercept | | -55.41 | 194.20 | 0.78 | 40 787.57 | 4054.22 | <0.001 | 0.4519 | 0.037 | <0.001 |
| Year | 2007[1] | Ref. | | | Ref. | | | Ref. | | |
| | 2008 | 29.94 | 3.76 | <0.001 | 908.57 | 91.39 | <0.001 | -0.0028 | 0.001 | 0.005 |
| | 2009 | 26.35 | 3.82 | <0.001 | 860.13 | 93.84 | <0.001 | 0.0482 | 0.001 | <0.001 |
| | 2010 | 13.65 | 3.86 | 0.04 | 603.47 | 94.95 | <0.001 | 0.0481 | 0.001 | <0.001 |
| | 2011 | -9.67 | 3.91 | 0.01 | 341.67 | 96.10 | <0.001 | 0.0630 | 0.001 | <0.001 |
| | 2012 | 1.64 | 3.96 | 0.68 | 949.91 | 97.53 | <0.001 | 0.0450 | 0.001 | <0.001 |
| | 2013 | 7.09 | 4.08 | 0.08 | 1 137.81 | 100.51 | <0.001 | 0.0357 | 0.001 | <0.001 |
| | 2014 | 21.52 | 4.10 | <0.001 | 1 337.70 | 100.80 | <0.001 | 0.0726 | 0.001 | <0.001 |
| | 2015 | 15.83 | 4.29 | <0.001 | 1 146.31 | 105.38 | <0.001 | 0.0581 | 0.001 | <0.001 |
| | 2016 | 36.07 | 4.48 | <0.001 | 1 303.78 | 110.48 | <0.001 | 0.0830 | 0.001 | <0.001 |
| Herd size[3] | Small[1] | Ref. | | | Ref. | | | Ref. | | |
| | Medium | -17.96 | 4.25 | <0.001 | -42.04 | 104.07 | 0.69 | -0.0176 | 0.001 | <0.001 |
| | Large | -35.15 | 5.32 | <0.001 | -365.30 | 129.57 | 0.005 | -0.0126 | 0.001 | <0.001 |
| | Very large | -52.62 | 9.89 | <0.001 | -1 050.12 | 240.97 | <0.001 | 0.0181 | 0.003 | <0.001 |
| Herd expansion log[2] | | -55.09 | 8.51 | <0.001 | -818.57 | 207.46 | <0.001 | -0.2203 | 0.002 | <0.001 |
| Heifer ratio | | -433.05 | 14.66 | <0.001 | -9 242.08 | 361.47 | <0.001 | 0.1269 | 0.004 | <0.001 |
| Birth first AI log (days) [2] | | | | | -1 458.69 | 350.24 | <0.001 | | | |
| Calve born first AI youngstock(%) | | | | | | | | -0.0001 | 0.000 | <0.001 |
| N AI youngstock | | | | | -142.51 | 70.47 | 0.04 | | | |
| Age first calving log (days) [2] | | 208.91 | 24.31 | <0.001 | -2 828.13 | 649.08 | <0.001 | | | |
| CI_log (days) [2] | | 179.73 | 21.82 | <0.001 | | | | -0.0433 | 0.006 | <0.001 |
| Calve born first AI (%) | | | | | 8.55 | 3.08 | 0.01 | | | |
| N AI | | | | | 517.14 | 116.73 | <0.001 | | | |
| High SCC (%) | | | | | | | | 0.0008 | 0.000 | <0.001 |
| Suspect subclinical ketosis | | -273.29 | 61.89 | <0.001 | -15 173.35 | 1 520.62 | <0.001 | 0.0984 | 0.017 | <0.001 |
| FPCM305 (kg) [4] | | -0.02 | 0.00 | <0.001 | 2.13 | 0.05 | <0.001 | | | |

[1] This group was used as reference category in the regression analyzes.

[2] Indictors were log transformed.

[3] Herd size was stratified by small, medium, large and very large based on the number of milking cows > = 50, 51–100, 101–200 and > = 201 cows respectively.

[4] FPCM305 = the average fat and protein corrected milk production in 305 days within a herd.

variance of average culling rate decreased with increasing herd size, the variance in the smallest sized group was extremely larger than in the other herd sized groups.

## Modelling results

In the three final mixed linear regression models, 8,11 and 8 herd performance indicators were significantly (P<0.001) associated with age of culled milking cows, lifetime milk production of culled cows and culling rate, respectively. Results of the three regression models are displayed in Table 3. The models only captured part of the variance that was observed among the herds over times as indicated by the relatively low marginal and conditional $R^2$. The marginal $R^2$ were 0.03, 0.08 and 0.23 in the model of culled milking cows, lifetime milk production of culled cows and culling rate. In addition, the conditional $R^2$ were 0.40, 0.40 and 0.49 respectively.

**Table 4. The estimated association of continuous herd performance indicators on age and lifetime milk production of culled milking cows and culling rate.** The coefficients of the continuous variables indicate the change in longevity variables resulting from a SD increase in mean values.

| | Mean | SD | Age culled cows (days) | | Lifetime production culled cows (kg) | | Culling rate | |
|---|---|---|---|---|---|---|---|---|
| | | | Regression coefficient | SE | Regression coefficient | SE | Regression coefficient | SE |
| Estimated value [1] | | | 2141.81 | 5.11 | 30858.10 | 124.63 | 0.206 | 0.0014 |
| Herd expansion log | 1.15 | 0.25 | -10.03 | 1.55 | -149.01 | 37.77 | -0.040 | 0.0004 |
| Heifer ratio | 0.24 | 0.06 | -26.53 | 0.90 | -566.14 | 22.14 | 0.008 | 0.0002 |
| Birth first AI log (days) | 491 | 54 | | | -142.28 | 34.16 | | |
| Calve born first AI youngstock(%) | 52.48 | 18.15 | | | | | -0.002 | 0.0003 |
| N AI youngstock | 1.9 | 0.5 | | | -67.43 | 33.34 | | |
| Age first calving log (days) | 792 | 48 | 11.45 | 1.33 | -155.00 | 35.57 | | |
| CI log (days) | 415 | 26 | 10.30 | 1.25 | | | -0.002 | 0.0003 |
| Calve born first AI (%) | 49.73 | 15.27 | | | 104.55 | 37.69 | | |
| N AI | 2.0 | 0.5 | | | 213.79 | 48.26 | | |
| High SCC(%) | 19.49 | 7.13 | | | | | 0.005 | 0.0004 |
| Suspect subclinical ketosis | 0.04 | 0.02 | -5.15 | 1.17 | -285.80 | 28.64 | 0.002 | 0.0003 |
| FPCM305(kg) [2] | 9 093 | 900 | -15.01 | 1.56 | 1 846.12 | 38.58 | | |

[1] Estimated value based on the reference levels of the categorical indicators and the mean values of continuous indicators. Categorical indicators (year and herd size) displayed the same value of Table 3.

[2] FPCM305 = the average fat and protein corrected milk production in 305 days within a herd.

Table 4 presents the coefficients based on the centered and scaled continuous variables. The regression results indicated that only four herd performance indictors shared a significant association with all three longevity variables. These indicators were herd size, herd expansion, heifer ratio and the proportion of cows with potential subclinical ketosis in a herd. The relevance of these herd performance indicators varied among the longevity variables as described below.

**Herd performance indicators associated with age of culled milking cows.** Age of culled milking cows in the very large (>201 cows), large size (101~200 cows) or medium size (51~100 cows) were successively significantly less than of small herds (<50 cows). The difference was 53, 35 and 18 days less, respectively. Additionally, herd expansion, heifer ratio, age at first calving, CI, ratio of number of suspected cows with subclinical ketosis and FPCM305 were also significantly (P<0.01) associated with age of culled milking cows. Heifer ratio had the strongest association with herd longevity. An increase in heifer ratio by one standard deviation (0.06) was associated with a decrease of 27 days in age of culled milking cows.

**Herd performance indictors associated with lifetime milk production of culled cows.** Compared with the model for age of culled milking cows, more reproduction performance indicators were included as significant explanatory variables (P<0.01) in the lifetime milk production model, such as number of AIs until conception, interval between birth and first AI (Table 4). Among all the significant herd indicators (P<0.01), lifetime milk production of culled cows was most sensitive for changes of FPCM305 and heifer ratio. One standard deviation (900 kg) increase of FPCM305 was associated with an increase of 1 846 kg in lifetime milk production. One standard deviation in heifer ratio (0.06) was associated with a reduction in lifetime milk production of culled cows by 576 kg (Table 4).

**Herd performance indicators associated with culling rate.** In the culling rate model, herd expansion was the most important associated variable: one standard deviation increase in

herd expansion (0.25) was associated with a decrease of culling rate by 0.04 (Table 4). Besides herd expansion, culling rate was strongly associated with herd size. In reference to the smallest sized herds, culling rate was lower in the medium to large herds, but higher in the very large sized herds. In addition, compared to the other herd sizes, smaller herds had a relatively large standard deviation in culling rate, indicating a larger heterogeneity among the herds in this herd size category. Although the coefficients of association between health indicators (i.e., the percentage of cows with high SCC and the ratio of cows with potential ketosis) and culling rate were relatively small, they reflected significant explanatory variables (P<0.01).

## Discussion

This study is based on census data from 10 719 commercial dairy herds in the Netherlands from the years 2007 to 2016 to explore the association of herd performance indicators with herd-level cattle longevity.

Due to changes in dairy farm management strategies associated with the milk quota abolishment in 2015, changes in average cattle longevity were expected over time. The three studied cattle longevity variables, however, remained relatively stable over the evaluated years. For instance, the annually averaged age of culled cows and lifetime milk production for years 2007 to 2016 ranged from 2 119 to 2 160 days and 30 523 to 31 683 kg, respectively (Table 2). The annually averaged culling rate (based on cows culled for slaughter) ranged from 0.21 to 0.26 (Table 2) over the evaluated years and was close to the previously described culling rate in the Netherlands of 0.25 [26]. Interestingly, the distributions of the three longevity variables were right-skewed, which indicates that there is a proportion of herds with a considerably higher cattle longevity. For instance, the average age of culled milking cows in a herd was 2 139 days, while the 95 percentile equaled 2 674 days, indicating a difference of almost 1.5 years (Table 1).

The regression results revealed that only four herd performance indicators were significantly associated with all three cattle longevity variables, i.e., heifer ratio, herd expansion, herd size and potential subclinical ketosis. It may not be surprising that heifer ratio is the most strongly associated with all three studied longevity variables (Table 4). The heifer ratio represents the relative number of replacement youngstock reared on a farm. In the Netherlands, most dairy farms rear their own replacement youngstock. The existence of abundant dairy heifers often results in shorter productive lifespans in herds of fixed sizes [27]. Culled cows are generally replaced by young stock to sustain or expand herd size. Considering the time lag between the decision on rearing a calve as a replacement heifer and the actual replacement of a dairy cow by this heifer, it is important for dairy farmers to plan the number of heifer calves to be reared as replacement animals. Herd expansion can be managed either by rearing more young stock, culling less cows, buying additional cows or a combination of these. Therefore, it is not surprising that in our models herd expansion was also associated with longevity. Similar to previous research [28], we found a negative association between herd size and longevity. The association between herd size and longevity may be caused by less personal attention and/ or a higher level of physiologic stress from increased mechanization with increasing herd size [28]. In addition, infectious disease may also lead to more culling in larger herds [29]. The percentage of cows that had potential subclinical ketosis was also associated with all longevity variables. Beside the direct impact of subclinical ketosis on culling risk, this health disorder is also known to be able to trigger other health problems such as displaced abomasum, cystic ovaries and mastitis [30], which may lead to earlier culling [31] and consequently lower longevity.

The significance of the associations between the remaining herd performance indicators varied with different longevity variables. However, the strength of these associations were generally weak. Remarkable to notice is the opposite association of age at first calving with age at

culling (positive) and lifetime milk production(negative). Our data showed that one SD increase in first calving age (48 days) was associated with an increase in age of culling of only 11.5 days, indicating a reduction in production lifespan. Since lifetime milk production is subjected by length of production lifespan and production intensity, the lifetime milk production will be reduced in the case of an increased calving age and an unaltered production intensity, explaining the negative association.

Low milk production, poor reproduction performance and occurrence of health disorders (peripartum health disorders, metabolic disorders, udder disorders) are commonly regarded as risk factors for culling of individual cows within a herd [32]. It is tempting to extrapolate the individual cow risk factors to the herd level. However, from a herd level perspective, the association of milk production level only strongly contributes to lifetime milk production. The associations with age of culling and culling rate are only mild or even insignificant, which contradict the often propagated public belief that intense milk production leads to short longevity and high culling rates. Similar reasoning can be carried out for the reproductive performance. A failure to conceive is a reason for individual cow culling [33]. When, as a management measure, farmers cull cows relatively quickly due to a failure to conceive, this will lead–on herd level—to a shorter CI. Many cow-level studies indicate strong correlations between CI and survival rate of cows [34]. Within in this study, the comparable associations of CI with the longevity features were mild at a herd level. One standard deviation increase in CI (26 days) was associated with a 10 days higher age of culled milking cows and a 0.002 lower culling rate only. In addition, since one standard deviation of CI (26 days) captures one insemination cycle in the Netherlands, the first association (higher age of culled milking cows) can be deduced from a notion that a higher tolerance for failure of conception or late first insemination may lead to less culling. Since CI excludes the group of animal that not get conception, herd reproduction performance can be better reflected by pregnancy rate. However, Information on pregnancy rate was lacking. In order to overcome this gap, other variables were selected as well, such as number of AI. Interestingly, other reproduction variables only displayed significant association with lifetime milk production of culled milking cows. Regarding the association of cattle health with longevity, the percentage of cows with high SCC in a herd, reflecting subclinical mastitis problems, was only significantly associated with a higher culling rate, although very weakly(small impact). Usually clinical mastitis is seen as the most important udder health parameter leading to removal of cows from the herd [35]. However, due to the lack of data, the relevance of this health disorder could not be studied.

Although a large observational cohort of Dutch commercial dairy farms was available for this study, the results revealed only limited associations between each of the evaluated longevity measures and herd performance indicators. Combined with the right-skewed distribution of each of the evaluated longevity measures, this finding indicates that there is potential for extending herd longevity without affecting the performance of the herd. Moreover, the relatively low marginal and conditional $R^2$ of the regression models disclose that only part of the observed variance in longevity among the herds over times was explained by the herd performance variables. This indicates that differences in longevity at herd level may be predominantly determined by other factors, like farmers' attitude and management style [4].

## Conclusion

In the Netherlands, the average longevity of dairy cattle at herd level, represented by age of culled milking cows, lifetime milk production and culling rate remained relatively constant over the years 2007 to 2016, although variance between herds was substantial. Four herd performance indicators (herd size, herd expansion, heifer ratio and suspects to subclinical ketosis)

were associated with all longevity aspects. However, the relevance of the herd performance indicators differed among the longevity variables, although most associations were rather weak. The absence of strong associations between herd performance and each herd-level longevity variables indicate that there is potential for extending herd longevity without affecting herd performance in terms of milk production, reproduction and health.

## Acknowledgments

The authors thank the Dutch Cooperative Cattle Improvement Organization CRV BV for providing the data. The authors also thank Dr. Bart van den Borne (chair group Business Economics, Wageningen University, Wageningen, the Netherlands) for help with the statistical modelling.

## Author Contributions

**Conceptualization:** Monique Mourits, Henk Hogeveen.

**Data curation:** Ruozhu Han, Monique Mourits, Wilma Steeneveld, Henk Hogeveen.

**Formal analysis:** Ruozhu Han.

**Methodology:** Ruozhu Han, Monique Mourits, Henk Hogeveen.

**Supervision:** Monique Mourits, Wilma Steeneveld, Henk Hogeveen.

**Writing – original draft:** Ruozhu Han.

**Writing – review & editing:** Monique Mourits, Henk Hogeveen.

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
