## [Decision Letter · Decision Letter 0]

12 Jul 2022

PONE-D-21-36421The association of herd performance indicators with cattle longevity: an empirical studyPLOS ONE

Dear Dr. han,

Thank you for submitting your manuscript to PLOS ONE. After careful consideration, we feel that it has merit but does not fully meet PLOS ONE’s publication criteria as it currently stands. Therefore, we invite you to submit a revised version of the manuscript that addresses the points raised during the review process.

Make the adjustments requested by the reviewers and send the paper back for us to finish.

Please underline the text to highlight the changes.

We look forward to receiving your revised manuscript.

Kind regards,

Julio Cesar de Souza, Ph.D.

Academic Editor

PLOS ONE

Journal Requirements:

2. Thank you for stating the following in your Competing Interests section: "There are no conflicts of interest to declare."

Additional Editor Comments (if provided):

Dear Sir,

Make the adjustments requested by the reviewers and send the paper back for us to finish.

Please underline the text to highlight the changes.

Best regards.

Reviewers' comments:

Reviewer's Responses to Questions

**Comments to the Author**

1. Is the manuscript technically sound, and do the data support the conclusions?

Reviewer #1: Yes

Reviewer #2: Yes

2. Has the statistical analysis been performed appropriately and rigorously? 

Reviewer #1: No

Reviewer #2: Yes

3. Have the authors made all data underlying the findings in their manuscript fully available?

Reviewer #1: No

Reviewer #2: Yes

4. Is the manuscript presented in an intelligible fashion and written in standard English?

Reviewer #1: Yes

Reviewer #2: Yes

5. Review Comments to the Author

Reviewer #1: This is a well conducted study. The statistics seem mostly sound, and the paper is mostly easy to follow. I have several comments that deserve attention. By line number.

Title: instead of cattle, better is dairy cow. Cattle can also be beef cows. Related to this, in your paper (various places) it appears that you do not include culling and mortality of heifers, so before females calve for the first time. Therefore the longevity estimates, starting from birth, are inflated. You should make clear that your metrics are conditional on females surviving to first calving.

L22: cull rate in your study is really annual cow cull rate. Please add annual, and make clear this is about cows being culled, not heifers. One could also calculate an annual heifer cull rate.

L27: An annual cull rate of 24% is quite low. Does this agree with data as published by from CRV or others with Dutch statistics? Quick math: if annual cull rate is 25%, then the productive lifetime is 4 years. In line 26, you write 5.8 years lifespan, so that would mean age at first calving is 1.8 years?

L47: I think longevity starts at birth?

L49: average lifespan “of a dairy cow” of 4.9 years is really the lowest average reported by farms, not for individual cows. Please clarify in the text.

L60: fewer replacement heifers. There are a few places where English could be improved a bit, but overall this is well written.

L98: A few means 2, you may have found more. I suggest you list them all, instead of just one [1].

L105: Explain what Dutch production conditions are, and how that would affect your results. Maybe expand in the Discussion.

L111: Are all your data herd level data, or do you also have individual cow records? The text seems to suggest you also analyzed individual cow records.

L116: you have 10 years worth of data, and 20 796 herds. So I was expecting more than 200,000 records. It seems you have fewer in table 1. Or is the 20 796 before editing? Best would be to show the number after edits.

Table 1: heifer ratio. Heifers are females from birth to first calving. I think you want to show the number of heifers on a farm compared to the milking cows? Farms that raise their heifers likely have 0.6 to 0.9 heifers per milking cow on the farm, not 0.24, because it takes 2 years to raise a heifer. The 0.24 is likely not correct.

Elsewhere in the table: “percentage of number” seems weird.

L146: should? I think you need to write what you did. Should is ambiguous.

L152: I see the 10719 herds are left over after editing, correct? But in L154-163 you are doing further editing so the final number of herds is less? You should have a final number of herds after all editing.

L154: individual means individual cows or herds?

L168: How do you know that that dataset you obtained exactly included the definitions laid out by Fetrow et al? It reads like Fetrow built the dataset, or you were able to edit the dataset such that you exactly follow the Fetrow definitions?

L179: This seems a different definition of heifer ratio than in table 1. This definition is very similar to your definition of cull rate if the herd stays at the same size, which is pretty much the case in your data. I think this definition of heifer ratio is counter intuitive: more interesting is the number of heifers on the farm vs the number of milking cows. Secondly, given your L179 definition is so similar to your cull rate definition, no wonder that you find that this heifer ratio is highly associated with cull rate.

L198: per cow per test day again suggests records per cow, not only per herd.

L211: clustering also suggests you had individual cow records. I am confused about what kind of records you had.

L228: probably not guarantee normality: data are seldom really normally distributed. I suggest you write more normally distributed. Overall, you did a nice job explaining how you checked your data and transformed if needed in these paragraphs.

L250: Did you use the default stopping criteria in R?

L260: Again I do not trust the 0.24 number.

L266 and later. You write about decreased or stayed the same or increased but did not statistically test these trends. You could use simple linear regression and see if your trend coefficient is significant. It is not correct to just report numerical differences without statistically testing.

Table 2: Can you add number of herds in a column? Was the cull rate really only 21% in 2008?

L285: You often write the word significant in the text, but better is to show the p-values and not use the word significant.

L307: seems text for the materials and methods. Also, how did you actually do this? Did you standardize your data by dividing by the SD, then looked at the regression coefficients?

Table 4: The top part of table 4 has the same values a the top part of table 3. That is not correct, you should not duplicate results. Why no p-values in table 4 but in table 3? The bottom part of table 4 does not have log values but the bottom part of table 3 does? These tables need some clean up.

L358: What kind of changes were expected? Can you at least give a direction (increase, decrease?)

L364: What do current cull rates statistics from say CRV say?

L374: Vague definition of heifer ratio, also not quite in line with earlier confusion about what heifer ratio actually is. For example, number first calvings does not include age at first calving, while the number of heifers on the farm depends on age at first calving.

L438: In your discussion of herd size, you see that greater herd size is associated with lower longevity. Yet Dutch “large” herds are still small compared to elsewhere. Can you speculated how longevity may be associated with herds that say have more than 1000 cows?

Reviewer #2: General comments

This is a well written manuscript that deals with questions of high importance for the development of dairy production and is clearly within the scope of the journal.

The study is well performed and contains a vast amount of relevant data and the presentation is clear and the discussion is relevant!

Specific comments

Line 19 Replace "is" with "was"

Line 38 Replace "times" with "time"

Line 50 Rephrase - change "far beyond" to something that does not indicate that you numbers are greater than the potential. Mayybe "below"

Line 65 Is there perhaps a reference to this?

Line 83 Replace "amount" with "number" and also "has" with "have"

Line 103 Replace "is" with "was"

Lines 116-117 I am not entirely convinced that it is appropriate to use the expression "Census" when it does not include data from all Dutch herds.

Lines 172-175 I do not ask for a change of herd size groups, but I would have considered Quartiles instead of your fixed limits leading to groups of rather different sizes.

Line 185 I do understand the use of CI (as it is used widely), however the variable is very dependent on management. Well woth discussing!

Line 235 I am not sure, but the reference should be for Table 2, right?

Line 395 Replace "was" with "were"

6. PLOS authors have the option to publish the peer review history of their article (what does this mean?). If published, this will include your full peer review and any attached files.

Reviewer #1: No

Reviewer #2: **Yes: **Nils Fall, Department of Clinical Scences, SLU

---

## [Author Response · Author response to Decision Letter 0]

30 Aug 2022

Reviewer 1#:

Title: instead of cattle, better is dairy cow. Cattle can also be beef cows. Related to this, in your paper (various places) it appears that you do not include culling and mortality of heifers, so before females calve for the first time. Therefore the longevity estimates, starting from birth, are inflated. You should make clear that your metrics are conditional on females surviving to first calving.

AU: We agree with the suggestion on the title, which has been adjusted accordingly. As indicated in line 177 and following, we only considered the lifespan of milking cows in our study. To further emphasise this specification we made some small modifications in line 176-183.

‘Cattle longevity variables. Three annually (production year) averaged cattle longevity variables were selected: age of culled milking cows(days), lifetime milk production of culled milking cows (kg), and culling rate. The number of culled cows represented all dairy cows, after first calving, removed from the milking herd for slaughter, salvage or death within a production year, according to the definition used by Fetrow et al [18]. Animals sold for production purposes to other dairy farms were excluded from this number. The culling rate was computed by dividing the number of culled milking cows by the number of milking cows in the same production year’

L22: cull rate in your study is really annual cow cull rate. Please add annual, and make clear this is about cows being culled, not heifers. One could also calculate an annual heifer cull rate.

AU: Adjusted accordingly throughout the manuscript and table. Information on heifers culled was lacking and, therefore, not considered in this study. 

L27: An annual cull rate of 24% is quite low. Does this agree with data as published by from CRV or others with Dutch statistics? Quick math: if annual cull rate is 25%, then the productive lifetime is 4 years. In line 26, you write 5.8 years lifespan, so that would mean age at first calving is 1.8 years?

AU: In our study, the definition of culling doesn’t include the cows that are removed from the herd to be sold for life (see line 181). As a result, the number of culled dairy cows is smaller than the number of dairy cows replaced, which is represented by the published rates of CRV. As such, a rate of 24% agrees with the findings of the study from Mohd Nor et al (2014) which showed that the Dutch annual cull rate under the same definition ranged from 23%-28%.( DOI: 10.1017/S0022029913000460)

L47: I think longevity starts at birth?

AU: As indicated in line 48, it is. 

L49: average lifespan “of a dairy cow” of 4.9 years is really the lowest average reported by farms, not for individual cows. Please clarify in the text.

AU: Adjusted accordingly in line 49. ‘ The lifespan of milking dairy cows in a herd is on average about 5.8 years ranging from 4.9 years to 7.1 years.’ 

L60: fewer replacement heifers. There are a few places where English could be improved a bit, but overall this is well written.

AU: ‘Less’ has been replaced by ‘fewer’ in line 60. 

L98: A few means 2, you may have found more. I suggest you list them all, instead of just one [1].

AU: Another citation has been added in line 100. (doi:10.1017/S1751731116001348)

L105: Explain what Dutch production conditions are, and how that would affect your results. Maybe expand in the Discussion.

AU: The “Dutch production conditions” was included to indicate that the data used was reflecting the Dutch situation; not to underline specific conditions. For this reason, we have modified lines 105 and 107. “The objective of this study was to explore the association of herd performance indicators with different herd-level dairy cow longevity aspects (age at culling, lifetime milk production and culling rate) using Dutch dairy herd performance data. The herd performance indicators included factors on milk production, youngstock rearing, reproduction and health performance as derived from Dutch dairy herd data registered during the period 2007-2016.”

L111: Are all your data herd level data, or do you also have individual cow records? The text seems to suggest you also analyzed individual cow records.

AU: All analyses are based on data at herd level. However, some of the raw data on health status was only available at individual level and was integrated at herd level prior to further analysis. To prevent further confusion ‘Herd-level data’ in line 111 has been replaced by ‘data’. For each subset database, more detailed information was given from lines 120 to 129.

L116: you have 10 years worth of data, and 20 796 herds. So I was expecting more than 200,000 records. It seems you have fewer in table 1. Or is the 20 796 before editing? Best would be to show the number after edits.

AU: 20,796 is the number of available herds from the raw database (or before editing). After the selection for representative farms, the final database consisted of 10,719 herds. This number has been displayed in line 162. 

Table 1: heifer ratio. Heifers are females from birth to first calving. I think you want to show the number of heifers on a farm compared to the milking cows? Farms that raise their heifers likely have 0.6 to 0.9 heifers per milking cow on the farm, not 0.24, because it takes 2 years to raise a heifer. The 0.24 is likely not correct. 

Elsewhere in the table: “percentage of number” seems weird.

AU: In our study, a heifer refers to a full grown replacement heifer or first calving cow (line 193). It is not indicating the number of youngstock.

‘percentage of number’ has been replaced by ‘percentage’ in Table 1.

L146: should? I think you need to write what you did. Should is ambiguous.

AU: ‘they should have had’ was replaced by ‘they had to have ’ in line 156. 

L152: I see the 10719 herds are left over after editing, correct? But in L154-163 you are doing further editing so the final number of herds is less? You should have a final number of herds after all editing.

AU: 10,719 is the final herd number. The further data editing (lines 164-173) is about editing (removing) the potential outlier entries in the herds’ records, hence, only removing specific annual herd data entries/records instead of the entire herd. 

L154: individual means individual cows or herds?

AU: It means specific herd data entries. To prevent confusion ‘individual’ has been removed from line 165.

L168: How do you know that that dataset you obtained exactly included the definitions laid out by Fetrow et al? It reads like Fetrow built the dataset, or you were able to edit the dataset such that you exactly follow the Fetrow definitions?

AU: This reference was to indicate that we adhered to the definition on culling number as set by Fetrow (number of milking cows removed from the herd, excluding the dairy cows that were sold for life); in order to avoid further ambiguity we have modified ‘according to the definition’ to ‘following the definition’ in line 180.

L179: This seems a different definition of heifer ratio than in table 1. This definition is very similar to your definition of cull rate if the herd stays at the same size, which is pretty much the case in your data. I think this definition of heifer ratio is counter intuitive: more interesting is the number of heifers on the farm vs the number of milking cows. Secondly, given your L179 definition is so similar to your cull rate definition, no wonder that you find that this heifer ratio is highly associated with cull rate.

AU: 

• It would certainly have been interesting to examine the association between the number of youngstock and the number of dairy cows, regrettably information on youngstock numbers was lacking.

• Heifer ratio was used to capture any changes in the size of the dairy herd (number of milking cows) or in the composition of the herd due to the upcoming milk quota abolishment in 2015. The definition of heifer (full grown replacement heifer or first calving cow) has been clarified throughout the manuscript and tables. 

• In order to avoid potential collinearity, the correlation coefficient between heifer ratio and cull rate was checked. The coefficient was less than 0.8, which was explained in line 257. 

L198: per cow per test day again suggests records per cow, not only per herd.

AU: The raw data on sub-clinical ketosis were available at cow level and aggregated to herd level for analysis. This clarification was adjusted in lines 213-216. ‘ Given this ratio per cow per test day, the herd ratio of cows with a high fat protein ratio was determined. The average ratio of suspected cows with subclinical ketosis within a herd was calculated as an average over all test days within one calendar year.’’

L211: clustering also suggests you had individual cow records. I am confused about what kind of records you had.

AU: The raw database consisted of 4 subsets. The subset related to the health status of the herd was recorded at cow level. A detailed description of each subset was given in lines 120-129. In order to do a herd level analysis, all recorded were integrated at herd level. We agree that the statement on line 227 creates confusion about the available data type. This line has been adjusted accordingly into ‘Herd as random effect was included in the linear mixed model to capture any other unobserved herd heterogeneity, such as specific herd management.’ 

L228: probably not guarantee normality: data are seldom really normally distributed. I suggest you write more normally distributed. Overall, you did a nice job explaining how you checked your data and transformed if needed in these paragraphs.

AU: Line 246 adjusted accordingly into ‘As a result, seven skewed variables were log transformed to be more normally distributed. ’

L250: Did you use the default stopping criteria in R?

AU: I didn’t using the default stopping criteria. We compared each BIC between each stepwise regression and stopped when the marginal change of BIC between steps increased dramatically. 

L260: Again I do not trust the 0.24 number.

AU: see earlier response; this rate is excluding dairy cows that were sold for life.

L266 and later. You write about decreased or stayed the same or increased but did not statistically test these trends. You could use simple linear regression and see if your trend coefficient is significant. It is not correct to just report numerical differences without statistically testing.

AU: we agree with this suggestion. The linear regression was applied to check longevity tendency with herd size. The results showed significant downward trend. The manuscript has been adjusted accordingly. ‘ Cattle longevity varied with herd size (Table 2). Average age of culled milking cows decreased with increasing herd size (trend coefficient p<0.001) ’ The following footnote has been added in Table 2. ’2the significant associations (p<0.001) tested by simple linear regression. ’

Table 2: Can you add number of herds in a column? Was the cull rate really only 21% in 2008?

Shows in Table 1

AU: 

• number of herds (10,719) did not change over years. The number of herds per herd size category was added in table footnote. Small-2269 herds; medium-8515 herds; large-5003 herds; very large-682 herds.

• 21% is the correct number; I also verified by other data sources like the Dutch Farmers Agricultural Data Network - FADN (https://www.agrimatie.nl/Binternet.aspx ) - see also response on comment L364

L285: You often write the word significant in the text, but better is to show the p-values and not use the word significant.

AU: Most of p-value are extremely small(e.g., 0.0004). Therefore, we indicated significant associations at the p-value less than or equal to 0.001, which was stated on line 311. This is in line with PLOS ONE’s requirement for p-value (https://journals.plos.org/plosone/s/submission-guidelines#loc-additional-information-requested-at-submission).

L307: seems text for the materials and methods. Also, how did you actually do this? Did you standardize your data by dividing by the SD, then looked at the regression coefficients?

AU: 

• The description on the standardization procedure was already included in materials and methods section on line 275. Therefore, the sentence has been deleted as the description there was indeed inappropriate.

• On line 275, we clarified that the continuous independent variables were centred and scaled before the regression analysis. To be more precise, centring is done by subtracting the mean value from the individual variable values, resulting in centred values. Scaling is done by dividing the centred variable values by the variable specific standard deviation.

Table 4: The top part of table 4 has the same values a the top part of table 3. That is not correct, you should not duplicate results. Why no p-values in table 4 but in table 3? The bottom part of table 4 does not have log values but the bottom part of table 3 does? These tables need some clean up.

AU: 

• We are agree the top parts of Table 3 and table 4 are duplicates. This part has been removed from Table 4. 

• The p-values in Table 4 are the same as in Table 3. Table 4 only shows the coefficients after centring and scaling the continuous variables. In order to avoid duplication, p-values were only displayed in Table 3.

• Thank you for indicating the “log” omission in Table 4. The Log label has been added in Table 4.

L358: What kind of changes were expected? Can you at least give a direction (increase, decrease?)

AU:. With the milk quota abolishment, production restrictions were relaxed, motivating farmers to expand their herd size by introducing more first calving heifers and/or by removing less milking cows (doi:10.1016/j.prevetmed.2021.105398). The final impact will depend on the farmer’s culling strategy. If the reduction in culling rate is evenly distributed over lactation numbers, then longevity is not altering. If the reduction is primarily in the higher lactation numbers, than longevity will decrease in the beginning and increase later on; if it is primarily in the earlier lactations it the average age of culled cows will increase. .

L364: What do current cull rates statistics from say CRV say?

AU: CRV is no longer reporting culling rates. Comparable information can be find in Farmers Agricultural Data Network (FADN), Wageningen Economic Research. The culling figures in FADN include also the number of dairy cows sold for live. Their cull rate is therefore a bit higher than the statistical results in our study. For verification, their cull rates are included (https://www.agrimatie.nl/Binternet.aspx).

 Year 2015 2014 2013 2012 2011 2010 2009 2008 2007

Number of sampled herds 297 285 290 294 297 288 284 275 259

Culling1 percentage (%) 21,7 25,8 24,1 25,2 28,8 26,9 26,4 23,7 23,1

 1 culling includes the cows that are removed from the herd for slaughter, salvage, death or sell.

L374: Vague definition of heifer ratio, also not quite in line with earlier confusion about what heifer ratio actually is. For example, number first calvings does not include age at first calving, while the number of heifers on the farm depends on age at first calving.

AU: Definition of heifer was corrected through manuscript – see earlier responses on heifer definition.

L438: In your discussion of herd size, you see that greater herd size is associated with lower longevity. Yet Dutch “large” herds are still small compared to elsewhere. Can you speculated how longevity may be associated with herds that say have more than 1000 cows?

AU:. It will be hard to predict the association between longevity and herds with more than 1000 cows, as it will strongly depend on the available resources (labour, land) and potential production restrictions (legislation). In general, the trend between longevity and herd size is expected to level off. For more extensive production systems the trend between longevity and herd size is expected to be less prominent.

Reviewer #2: 

Line 19 Replace "is" with "was"

AU: Adjusted accordingly.

Line 38 Replace "times" with "time"

AU: Adjusted accordingly.

Line 50 Rephrase - change "far beyond" to something that does not indicate that you numbers are greater than the potential. Mayybe "below"

AU: . Adjusted accordingly.

Line 65 Is there perhaps a reference to this?

AU: The reference ‘Longevity as an Animal Welfare Issue Applied

to the Case of Foot Disorders in Dairy Cattle’ (DOI: 10.1007/s10806-012-9376-0) has been added.

Line 83 Replace "amount" with "number" and also "has" with "have"

AU: Adjusted accordingly.

Line 103 Replace "is" with "was"

AU: Adjusted accordingly.

Lines 116-117 I am not entirely convinced that it is appropriate to use the expression "Census" when it does not include data from all Dutch herds.

AU: We agree. The ‘census’ statement has been removed.

Lines 172-175 I do not ask for a change of herd size groups, but I would have considered Quartiles instead of your fixed limits leading to groups of rather different sizes.

AU: Thank you for your remark. The fixed limits of 50, 100 and 200 cows are in line with the herd size from the 25(62 cows),75(107 cows) and 95(173 cows) quartiles. 

Line 185 I do understand the use of CI (as it is used widely), however the variable is very dependent on management. Well woth discussing!

AU: we realized the limitation of using CI to reflect herd reproduction performance. especially it excludes the group of animal that not get conception. Pregnancy rate, ideally, is the gold standard for assessing fertility of cows and heifers. Information on pregnancy rate was lacking, hence, not considered in this study. In order to overcome this gap, other variables were selected as well, such as number of AI. Manuscript adjusted accordingly.

Line 235 I am not sure, but the reference should be for Table 2, right?

AU: No, it should relate to Table 1. In the foot note below Table 1.

Line 395 Replace "was" with "were"

AU: Adjusted accordingly.

---

## [Decision Letter · Decision Letter 1]

14 Nov 2022

The association of herd performance indicators with dairy cow longevity: an empirical study

PONE-D-21-36421R1

Dear Dr. han,

We’re pleased to inform you that your manuscript has been judged scientifically suitable for publication and will be formally accepted for publication once it meets all outstanding technical requirements.

Kind regards,

Angel Abuelo, DVM, MRes, MSc, PhD, DABVP (Dairy), DECBHM

Academic Editor

PLOS ONE

Additional Editor Comments (optional):

Reviewers' comments:

Reviewer's Responses to Questions

**Comments to the Author**

1. If the authors have adequately addressed your comments raised in a previous round of review and you feel that this manuscript is now acceptable for publication, you may indicate that here to bypass the “Comments to the Author” section, enter your conflict of interest statement in the “Confidential to Editor” section, and submit your "Accept" recommendation.

Reviewer #1: All comments have been addressed

Reviewer #2: All comments have been addressed

2. Is the manuscript technically sound, and do the data support the conclusions?

Reviewer #1: Yes

Reviewer #2: Yes

3. Has the statistical analysis been performed appropriately and rigorously? 

Reviewer #1: Yes

Reviewer #2: Yes

4. Have the authors made all data underlying the findings in their manuscript fully available?

Reviewer #1: No

Reviewer #2: Yes

5. Is the manuscript presented in an intelligible fashion and written in standard English?

Reviewer #1: Yes

Reviewer #2: Yes

6. Review Comments to the Author

Reviewer #1: (No Response)

Reviewer #2: Thank you for a good review and a good paper!

All my comments have been adressed in a very good way.

7. PLOS authors have the option to publish the peer review history of their article (what does this mean?). If published, this will include your full peer review and any attached files.

Reviewer #1: No

Reviewer #2: No

---

## [Editor Report · Acceptance letter]

14 Dec 2022

PONE-D-21-36421R1 

The association of herd performance indicators with dairy cow longevity: an empirical study 

Dear Dr. Han:

I'm pleased to inform you that your manuscript has been deemed suitable for publication in PLOS ONE. Congratulations! Your manuscript is now with our production department. 

Kind regards, 

on behalf of

Dr. Angel Abuelo 

Academic Editor

PLOS ONE